# Single Nucleotide Polymorphisms as Biomarker Predictors of Oral Mucositis Severity in Head and Neck Cancer Patients Submitted to Combined Radiation Therapy and Chemotherapy: A Systematic Review

**DOI:** 10.3390/cancers16050949

**Published:** 2024-02-27

**Authors:** Ronaldo Cavalieri, Harley Francisco de Oliveira, Thais Louvain de Souza, Milton Masahiko Kanashiro

**Affiliations:** 1Centro de Biociências e Biotecnologia, Universidade Estadual do Norte Fluminense Darcy Ribeiro, Campos dos Goytacazes 28013-602, Brazil; ronaldocavalieri@gmail.com; 2Centro de Radioterapia, Grupo OncoBeda, Campos dos Goytacazes 28010-140, Brazil; 3Hospital das Clínicas da Faculdade de Medicina de Ribeirão Preto, Universidade de São Paulo, Ribeirão Preto 14015-010, Brazil; harley@fmrp.usp.br; 4Faculdade de Medicina de Campos, Campos dos Goytacazes, Rio de Janeiro 28035-581, Brazil; thaislsouza@gmail.com

**Keywords:** head and neck cancer, mucositis, chemoradiotherapy, SNPs

## Abstract

**Simple Summary:**

Single Nucleotide Polymorphisms (SNPs) are the most common type of genetic variation found in an individual’s DNA sequences. SNPs can occur in both coding and non-coding regions of the genome and can affect gene expression, protein function, and disease susceptibility. In this systematic review, we evaluate the potential of SNPs as biomarkers in the assessment of oral mucositis (OM) severity in head and neck cancer (HNC) patients treated with concomitant chemoradiation (CRT). This systematic review demonstrates that SNPs in different biological pathways have the potential to be biomarkers and to function as predictors of patients who will develop severe mucositis. The most studied biological pathway was that involving DNA damage repair genes. However, considering the involvement of cytokines and inflammatory pathways in the pathophysiology of mucositis, we believe that this is a promising area in the study of the prediction of this condition. We conclude that SNPs can be used as possible biomarkers for the assessment of OM intensity in HNC patients, and further research is needed to explore the potential of SNPs in personalized medicine for HNC treatment.

**Abstract:**

Single Nucleotide Polymorphisms (SNPs) are the most common type of genetic variation found in an individual’s DNA sequences. SNPs can occur in both coding and non-coding regions of the genome and can affect gene expression, protein function, and disease susceptibility. In this systematic review, we evaluate the potential of SNPs as biomarkers in the assessment of oral mucositis (OM) severity in head and neck cancer (HNC) patients treated with concomitant chemoradiation (CRT). The study selection process involved screening 66 articles from different platforms, and after removing duplicates and excluding articles that did not meet the eligibility criteria, 23 articles were included for full-text evaluation. Among them, genes from several pathways were analyzed. The DNA damage repair pathways had the highest number of genes studied. The most frequently analyzed gene was *XRCC1*. The proinflammatory cytokine pathways evaluated were TNF, with three articles, and NF-κB, with one article. Most included studies showed a potential association between certain SNPs and high-grade mucositis. We conclude that SNPs can be used as possible biomarkers for the assessment of OM intensity in HNC patients, and further research is needed to explore the potential of SNPs in personalized medicine for HNC treatment.

## 1. Introduction

Head and neck cancer (HNC) represents a heterogeneous group of malignancies located in an anatomical region containing delicate structures with close relationships to each other and involved in critical functions such as feeding and breathing. In addition, the natural history of the disease and the adverse effects of treatment can directly impact the patient’s appearance, verbal expression, and social interaction [1]. It originates from epithelial or glandular tissue lining the aero-digestive tract, and it can involve the oral cavity, pharynx, nasal cavity, larynx, thyroid, and salivary glands. HNC is a relatively common cancer worldwide, with approximately 950,000 new cases and 470,000 deaths per year across all subsites, and the global incidence is estimated to increase by 34% by 2030 [2]. Until the early 1990s, local curative treatment was primarily based on surgery with or without postoperative radiotherapy (RT). Later on, several publications started to demonstrate the possibility of organ preservation without loss of overall survival in selected groups of patients treated with CRT, making combined treatment a viable option [3,4]. However, conservative treatment with concurrent chemotherapy (CT) has altered and intensified some of the already known toxicity patterns of RT [5].

Mucositis is a frequent and acute complication, occurring regularly in patients receiving CRT. The patient may develop, to varying degrees, dysphagia, odynophagia, dehydration, and weight loss, and it is often necessary to temporarily interrupt RT or reduce the dose of CT, affecting outcomes and increasing frequency, duration, and costs of hospitalization and antibiotic therapy [5]. Worsening severity of mucositis is related to declining levels of quality of life [6]. Classically, it was described exclusively as a direct and local effect of radiation damage to the DNA of basal layer cells, influencing their multiplication and replacement of mucosal surface cells. Radiation-induced mucositis was classically recognized as an “outside-in” process. The concept of mucosal injury as a biological and multifactorial process is relatively new [6,7].

In 2004, Sonis et al. published a study describing aspects of the development of mucositis as a dynamic process involving the activation of different pathways, described in a model of distinct phases [7]. The complexity of its pathogenesis, with description of distinct biological pathways and associated systemic factors, has been increasingly explored [8].

Previous data have reported a possible association between genetic polymorphisms in different pathways and the development of mucositis, acting as potential biomarkers of severity [9]. The most common type of polymorphism is one involving a single nucleotide, called single nucleotide polymorphism (SNP) [10].

The early identification, through the analysis of molecular aspects, of patients who are possibly more likely to develop mucositis, as well as its intensity, represents a promising area and the possibility of a new approach to this frequent complication. Individuals identified early on with a greater probability of developing this debilitating toxicity can be approached differently with the preemptive implementation of preventive measures and aggressive supportive care before and during radiotherapy. The purpose of this systematic review is to identify studies that evaluated SNPs in different pathways as predictive biomarkers of mucositis intensity in head and neck cancer patients submitted to combined radiotherapy and chemotherapy.

## 2. Methods

This systematic review was performed according to the Preferred Reporting Items for Systematic Reviews and Meta-Analyses (PRISMA) checklist [11] and registered as 10.17605/OSF.IO/VQJPA.

### 2.1. Eligibility Criteria

Articles were included that addressed the relationship between SNPs in different biological pathways as biomarkers of the severity of oral mucositis (OM) in HNC patients receiving RT or CRT. The treatments had to be curative and RT could be used as a radical or postoperative treatment.

Articles were excluded because of the following reasons: evaluated patients with other types of cancer; HNC treatment without CRT; no correlation of the SNP with the severity of mucositis; reviews, letters, personal opinions, book chapters, and conference abstracts; association between biomarkers and OM in experimental studies (clinical trials, in vitro or in vivo animal studies); and language restrictions.

### 2.2. Information Sources and Search Strategies

An exhaustive search was carried out in the main databases used as research tools—PubMed, LILACS, Science Direct, and Cochrane. A search was also carried out on Google Scholar in order to detect any publications in addition to those already listed. Search strategies included the use of the following terms: “head and neck cancer OR head and neck carcinoma” AND “radiation therapy OR radiotherapy OR chemotherapy OR chemoradiation” AND “mucositis OR oral mucositis” AND “single nucleotide polymorphisms OR SNPs OR SNP OR gene polymorphism”.

Articles published between January 2009 and September 2022 were included for analysis. All duplicate articles were removed. In addition, the references of the included articles were reviewed to detect publications that were not identified in the search process.

### 2.3. Study Selection and Data Collection Process

In the first phase, the articles were listed in the databases and all the titles and abstracts were reviewed by two independent authors (RC and HO), who made the primary selection. The second phase consisted of reading the full text and retrieving those that met the inclusion criteria. This stage was also carried out by the same two authors (RC and HO). In the event of disagreement, a third author (MK) decided whether or not to include the reference. The references of the articles were reviewed by the first author (RC).

For all the articles included, the following information was recorded: authors, year of publication, country, number of patients, RT dose, treatment modalities, pathways of the polymorphisms studied, type of study, and main conclusions.

### 2.4. Study Risk of Bias and Quality Assessment

The quality and possible sources of bias of the selected studies were assessed using the Newcastle Ottawa Scale (NOS) tool, specific to observational studies. Three quality parameters (selection, comparability, and results), divided into eight specific items, were assessed. Each item on the scale was scored out of one point, except comparability, which was scored out of two points. The maximum for each study was nine, and references with less than five points were identified as representing a considerable risk of bias [12].

Two authors independently verified the parameters (RC and TL). A third author (HO) decided in case of disagreement.

## 3. Results

### 3.1. Study Selection

After the terms were restricted, we found a total of 66 articles on different platforms. Duplicated articles were removed, leaving 53 articles. The titles and abstracts of all of them were subsequently analyzed comprehensively. Twenty-seven articles were selected for full-text review. Three additional articles were found from the references of the selected articles. Finally, 23 articles were selected for final analysis (Figure 1 From PRISMA, http://www.prisma-statement.org/?AspxAutoDetectCookieSupport=1 (accessed on 30 October 2023)).

### 3.2. Study Characteristics

The countries where the studies were conducted were China (10), Poland (04), India (03), Japan (01), Belgium (01), Italy (01), Spain (01), France (01), and the USA (01).

All articles were published in English, between 2009 and 2021.

The total of individuals from the published articles was 4977. Genome-wide association studies contributed to most of them. Sample size ranged from 24 to 1467 patients with HNC. All patients received RT with or without CT. Two articles included surgery as an option.

All articles used genomic DNA extracted from blood samples, except one that used tumor samples stored in liquid nitrogen from diagnostic biopsies.

Table 1 summarizes the main characteristics and findings of the selected articles.

### 3.3. Synthesis of Results

#### 3.3.1. Mucositis SNP-Associated Non-Proinflammatory Mediator-Regulated Genes

Among the 23 included articles, genes from several pathways were analyzed.

Eight studies assessed the relationship between acute toxicities and polymorphisms in genes related to DNA damage repair (Figure 1). The most frequently analyzed gene was XRCC1 (Figure 1 and Figure 2). The DNA damage repair pathways had the highest number of genes studied (Figure 3). One article found no substantial association between SNPs in DNA repair genes and mucositis [13].

Three groups analyzed and established the relationship of an SNP at codon 399 of the *XRCC1* gene [23,30,34]. Chen et al. observed no significant difference in the severity of acute OM damage during RT between patients with different genotypes [19]. However, in one article there was a significant correlation (*p* = 0.011) [24] and in another a marginal correlation (*p* = 0.065) [15].

Three articles published results on the relationship between the Arg194Trp polymorphism in the *XRCC1* gene and clinical outcomes in patients with head and neck tumors. Li et al. showed the polymorphism was associated with a decreased incidence of grade 3 acute OM compared with the Arg/Arg allele, but with no statistical significance [15]. The other two articles concluded that the presence of SNP was significantly related to mucositis, *p* = 0.023 [25] and *p* = 0.01 [34].

Seven SNPs of four genes of the Wnt/β-catenin pathway were investigated. The rs454886 polymorphism of the adenomatous polyposis coli gene was correlated with grade 3–4 OM (*p* = 0.045) [17].

Additional studies have found a relationship between mucositis and SNPs in non-coding RNA pathways, in genes encoding ghrelin [26], in the *ABCC1* gene pathway [28], and in autophagy-related genes [29].

Four studies performed a genome-wide association analysis. They were included as they found a link between the occurrence of different polymorphisms and the potential risk of developing mucositis. One study was on the *CCHCR1* gene [35], the others on the *TNKSK* [33], *ZNF24* [21], and *RB1* [23] pathways.

#### 3.3.2. Mucositis SNP-Associated Proinflammatory Mediator-Regulated Genes

Previous data have established the association of inflammatory cytokines and their pathways in the development, maintenance, and intensity of acute mucositis. Clinical inflammatory conditions are preceded by increased levels of these cytokines. Four articles evaluated the relationship between SNPs in inflammatory cytokine pathways and mucositis (Figure 2).

The cytokine pathways evaluated were TNF, with three articles, and NF-κB, with one article. The TNF receptor, TNFR1, when activated, regulates an apoptotic pathway and may be involved in the development of mucositis. Two articles assessed SNPs in the promoter region of the gene encoding the receptor protein, TNFRSF1A, and showed a significant increase in the likelihood of grade 3 mucositis in the last weeks of RT in CC, TT, or GT genotype carriers [27,31]. The other article addressing the TNF pathway assessed the relationship between the SNP (-1211 T>C, rs1799964) in the *TNF-α* gene itself and the occurrence and intensity of OM [32].

Guo et al. chose five SNPs in the NF-κB pathway to determine whether they related to the likelihood of acute toxicity. There was no relationship between the polymorphisms and mucositis. However, in the same article, the authors also evaluated SNPs in cyclins, cell cycle regulatory proteins. CCND1 rs9344 was associated with grade 3–4 radiation-induced acute OM [20].

### 3.4. Study Risk of Bias and Quality Assessment

All the references were individually assessed using the NOS tool, with a maximum score of nine. A total of 9 articles (39.1%) obtained a score of six, 11 (47.8%) a score of five. Three (13%) articles scored four, and only these were considered to be of fair quality (Appendix A).

## 4. Discussion

In recent years, much progress has been made in understanding the molecular pathways associated with the pathophysiology of diseases, as well as treatments and their adverse effects. In oncology, it is no different; several signaling pathways have been described, and the personalization of the diagnostic and therapeutic approach is a remarkable advance. The treatment of HNC based on RT combined with CT represents a possible cure for certain diseases that previously did not have curative surgical approaches, in addition to offering a chance of organ preservation in an anatomical region very sensitive to these changes [4]. One of the most common and debilitating toxicities, with the potential to compromise therapeutic outcomes related to this strategy, is mucositis [6].

Mucositis is a common and often debilitating side effect of cancer treatment that involves damage to the mucous membranes lining the gastrointestinal tract, particularly in the oral and oropharyngeal regions. It can be caused by CT, radiation therapy, or a combination of both, and can lead to a range of clinical and symptomatic manifestations, including erythema, ulceration, pain, and difficulty eating and drinking [36]. Mucositis can also increase the risk of infection and other complications [37]

Historically, mucositis was seen as a result of a direct action of RT or CT on the cells of the basal epithelial layer. However, this simple picture could not explain the complex interactions that started to be described in the submucosal layer. The concept that mucosal injury is a biological and multifactorial process is relatively new [6,7].

In 2004, Sonis et al. published a study describing the aspects of mucositis development as a dynamic process divided into phases, initially involving the generation of free radicals by direct action of RT and/or CT [7]. Their research showed that local damage triggers an inflammatory response, activating several biological pathways, including nuclear factor kappa-B (NF-kB). This transcription factor, which can also be directly activated by RT and/or CT, induces gene expression of proinflammatory cytokines such as interleukin-6 (IL-6), interleukin-1β (IL-1β), and tumor necrosis factor α (TNF-α), which are apparently increased in mucositis. TNF-α may engage in a positive feedback mechanism with NF-kB, increasing its concentration, and eventually producing a massive release of cytokines (proinflammatory cytokine storm), amplifying the process. The clinical condition then develops with the onset of disease-related signs and symptoms, such as ulcers. At this point, the patient is more susceptible to infections, which can prolong mucositis as the activation of macrophages can release more inflammatory cytokines [7].

Recent advances in molecular and cell biology have provided targets for mechanism-based interventions to prevent and treat mucositis. Normando et al. evaluated several biomarkers for their association with the development of OM, including epidermal growth factor, C-reactive protein, genetic polymorphisms, tumor necrosis factor alpha (TNF-α), erythrocyte sedimentation rate, growth factors, acute-phase inflammatory markers, cytokines, general proteins, plasma antioxidants, apoptotic proteins, and cells. The meta-analysis showed an expression of polymorphisms in XRCC1, XRCC3, and RAD51 genes, as well as an expression of protein biomarkers, in patients with an increased risk of developing OM. However, the effectiveness of these biomarkers in predicting mucositis risk and guiding treatment decisions is still under investigation [9].

Genetic polymorphisms can influence gene expression by affecting various regulatory elements within the genome. Among genetic polymorphisms, SNPs are the most common type of genetic variation in the human genome [10]. Several groups have studied the relationship between SNPs and different clinical outcomes. The development of the disease, the different responses to the therapies implemented, and the development of more intense adverse effects to the treatments may be related to the occurrence of specific SNPs. Searching for and establishing this relationship represents an area of study that could alter and personalize approaches to a wide range of conditions in different areas.

Recently, several studies have explored the relationship between SNPs and susceptibility to mucositis in cancer patients. In this systematic review, we analyzed the non-proinflammatory and proinflammatory genes regulated by SNPs in HNC patients undergoing RT and CT.

Proinflammatory cytokines play a decisive role in the development of mucositis [38]. The presence of polymorphisms in genes that code for proinflammatory proteins can lead to alterations, causing an increase in the release of cytokines and intensification of mucositis. Despite the preponderant role of inflammatory cytokines in the development of mucositis, to date few studies have been published demonstrating a relationship between SNPs in these pathways and this adverse effect. In our review, four publications studied SNPs in these pathways.

In the case of TNF, the process of apoptosis is regulated through the TNFR1 receptor. SNPs in the gene that codes for the receptor can alter its expression and function and be linked to adverse events [39]. Brzozowska et al. showed that the presence of the T allele in the TNFRSF1A gene is associated with an increased risk of manifestation of grade 3 OM in patients with HNC undergoing RT. The study found that patients with TT or GT genotypes demonstrated a higher risk of grade 3 OM manifestation at week five of RTH compared to GG carriers. The results indicate an association between the TNFRSF1A gene SNP (rs4149570) and the risk of more severe RT-related OM in patients with HNC [27].

In their work, Mlak et al. (2020) showed that the CC genotype of the TNFRSF1 A gene is associated with a more severe course of OM [31]. The SNP identified in this study is located in the regulatory region of the TNFRSF1 A gene. Specifically, it is the SNP (rs767455) that has been found to be involved in the regulation of TNFR1 protein expression and may potentially modulate the risk of OM in HNC patients treated with RT, and it is an independent prognostic factor of poor overall survival in HNC patients undergoing intensity-modulated radiation therapy. The TT genotype, in contrast, had an inverse protective effect. However, due to the limitations of the study, further research is needed to confirm these results.

Another inflammatory pathway studied was the NFKB transcription factor, which when activated causes up-regulation of genes that result in increased cytokine production. The κB kinase inhibitor gene IKBKB rs12676482 was related to grade 3–4 radiation-induced acute myelosuppression, but not to mucositis [18].

In the studies that have evaluated SNPs in non-inflammatory pathways, the most investigated have been alterations in DNA damage repair genes. DNA damage repair is a mechanism which is related to radiosensitivity in both tumor response and treatment-related adverse effects [40]. Normal tissues depend on this mechanism to recover between RT sessions from damage caused by RT, and alterations in this mechanism can exacerbate toxicities, increasing variability between individuals receiving the same treatment. The included articles evaluated polymorphisms in candidate genes previously related to radiosensitivity. The most common gene studied was XRCC1, which is associated with the ability to detect and repair three of the most common DNA errors: single-strand breaks; double-strand breaks, which have greater biological significance; and base excision repair [40]. The most frequently searched SNP was c1196A>G p.Gln399Arg [15,19,24].

The risk of bias and the quality of the articles were assessed using the NOS tool, which establishes points for pre-established criteria [12]. The articles generally have a similar design, reducing the possibility of bias. In general, the articles were considered to be of good quality and to have a low risk of bias, since most of them scored five or more. In all the publications, the control groups were not patients who had not been exposed to the treatment, but patients with milder toxicities, which is a characteristic of the study design. However, according to the NOS scale, it is not possible to score them in this respect. Another issue that is not specified in most references is the demonstration that the main outcome, mucositis, is not present at the start of the studies. It is understood that mucositis was not present before the start of RT, but when this is not specified, it cannot be scored using the scale.

This systematic review demonstrates that SNPs in different biological pathways have the potential to be biomarkers and to function as predictors of patients who will develop severe mucositis. The most studied biological pathway was that involving DNA damage repair genes. However, considering the involvement of cytokines and inflammatory pathways in the pathophysiology of mucositis, we believe that this is a promising area in the study of the prediction of this condition. As studies collect more robust data on biomarkers in different pathways, we will be closer to developing a more personalized treatment strategy for a clinical condition that is costly for healthcare systems and so debilitating for patients who already have a very challenging type of cancer.

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
