# Peer review of "Single Nucleotide Polymorphisms as Biomarker Predictors of Oral Mucositis Severity in Head and Neck Cancer Patients Submitted to Combined Radiation Therapy and Chemotherapy: A Systematic Review"

_cancers, 2024, doi:10.3390/cancers16050949_

Round 1

Reviewer 1 Report

Comments and Suggestions for Authors

The current version is not suitable for publication. This is claimed to be the review article but I did not understand why the authors have introduce the article format including such sections e.g.; Methods, Results, etc. Secondly, more literatures should be included. Also, the topic is not significant to be published.

Comments on the Quality of English Language

Should be substantially improved.

Author Response

Comments and Suggestions for Authors

R1: The current version is not suitable for publication. This is claimed to be the review article but I did not understand why the authors have introduce the article format including such sections e.g.; Methods, Results, etc. Secondly, more literatures should be included. Also, the topic is not significant to be published.

Comments on the Quality of English Language

Should be substantially improved.

R: We thank the reviewer for his willingness to read our manuscript, but we clarify that our systematic review proposal strictly followed the PRISMA methodological guidelines at http://www.prisma-statement.org/?AspxAutoDetectCookieSupport=1

The entire manuscript was reviewed by a professional English proofreader

Reviewer 2 Report

Comments and Suggestions for Authors

1. The title must be more informative. SNPs function as a biomarker or as multiple biomarkers? assessment of what? diagnostics? prognostics? disease progression? 

2. Abstract:

- must also be more informative. biomarker or biomarkers? of what? diagnosctics? prognostics? disease progression? The expression "influence of SNPs" does not seem very approppriate. Perhaps the potential of SNPs to be used as biomarkers? 

- I consider that it would be more important to mention the results observed than to mention the amount of consideres/included articles...

3. Introduction:

- The first paragraph lacks references.

- The first three paragraphs must be merged and the information connected. it is the same topic, so there is no need to have three separate paragraphs.

- the forth and fifth paragraph must be transform in one as it approaches the same topic: mucositis.

 - the fifth paragraph must contain references.

- The end of the introduction must contain the relevance of this manuscript, regarding to the importance of SNPs specifically in oral mucositis, and not in general terms.

4. Methods

- It is only with in the elibigibility criteria section that the reader fully understand the goal of this study. The title, abstract and end of the introduction must contain this information in a precise and direct manner!

 - the search terms must be included

5. Results

- The Flow diagram of the studies selections is not numbered. Please do so and mention it in the tex.

- In “Flow diagram of the studies selections”: please verify if you used the same font as  the manuscript, correct “indentified” to “identified”, “screened from database” to “screened from databases”.

- Google scholar is never mentioned in the methodology….

- Table 1: Please explain the rational of include the country and dose of treatment

- Table 1: I believe that the main conclusions must be summarized and presented in topics rather than text. This column must appear sooner in the table and before the “signalling pathway” column.

- Table 1: rename the column “signalling pathway”, because it actually refers to “genes”.

- include references at section 3.3.2.

- figures 1, 2 and 3 must be improved: font (same as the manuscript), colour (black legend), remove the superior legend. Remove “series 1 and 2”.

- figures 1-3 must be mentioned in the text.

6. Discussion

- Please include references in ALL information that is not yours! Otherwise, is considered plagiarism.

- Discussion must be re-written. I believe that after performing the changes suggested in these comments and after digesting further the results, it would be easier to write the discussion.

Author Response

Reviewer 2

R: We thank the reviewer for the careful and critical reading of our manuscript. The alterations are highlighted in yellow in the manuscript.

Comments and Suggestions for Authors

  1. The title must be more informative. SNPs function as a biomarker or as multiple biomarkers? assessment of what? diagnostics? prognostics? disease progression? 

R: We changed the title to “Single nucleotide polymorphisms as biomarker predictors of oral mucositis severity in head and neck cancer patients submitted to combined radiation therapy and chemotherapy: A systematic review”

  1. Abstract:

- must also be more informative. biomarker or biomarkers? of what? diagnosctics? prognostics? disease progression? The expression "influence of SNPs" does not seem very approppriate. Perhaps the potential of SNPs to be used as biomarkers? 

R: We agreed with the reviewer's suggestions and made the suggested changes and they are highlighted in yellow.

- I consider that it would be more important to mention the results observed than to mention the amount of consideres/included articles...

R: We appreciate the reviewer's suggestions as they significantly improved the manuscript’s abstract. All changes are highlighted in yellow.

  1. Introduction:

- The first paragraph lacks references.

R: Reference added (1)

- The first three paragraphs must be merged and the information connected. it is the same topic, so there is no need to have three separate paragraphs.

R: Reviewer's suggestion met. We put the three paragraphs together.

- the forth and fifth paragraph must be transform in one as it approaches the same topic: mucositis.

R: We agreed with the reviewer's suggestion and merged the two paragraphs.

 - the fifth paragraph must contain references.

R: Reference added (6, 7)

- The end of the introduction must contain the relevance of this manuscript, regarding to the importance of SNPs specifically in oral mucositis, and not in general terms.

R: We appreciate the reviewer's suggestions as they significantly improved our manuscript. All changes made are highlighted in yellow.

  1. Methods

- It is only with in the elibigibility criteria section that the reader fully understand the goal of this study. The title, abstract and end of the introduction must contain this information in a precise and direct manner! 

R: We appreciate the reviewer's suggestions, all of which were accepted and the changes are highlighted in yellow.

- the search terms must be included

R: The terms searched were included in the method “Search strategies included the use of the following terms: “head and neck cancer OR head and neck carcinoma” AND “radiation therapy OR radiotherapy OR chemotherapy OR chemoradiaton” AND “mucositis OR oral mucositis” AND “single nucleotide polymorphisms OR SNPs OR SNP OR gene polymorphism”.

  1. Results

- The Flow diagram of the studies selections is not numbered. Please do so and mention it in the tex.

R: We made changes to the flow diagram and changes to the text were made according to the reviewer's suggestions.

- In “Flow diagram of the studies selections”: please verify if you used the same font as the manuscript, correct “indentified” to “identified”, “screened from database” to “screened from databases”.

R: All suggested corrections were made in the flow diagram.

- Google scholar is never mentioned in the methodology….

R: The suggested correction was made in the methodology

- Table 1: Please explain the rational of include the country and dose of treatment

R: We decided to include the country of origin of the study in order to draw attention to the geographical region of the population included in each publication, as it involves the analysis of genetic polymorphisms.

R: The dose of radiotherapy was reported to draw attention to the fact that all patients underwent treatment with curative rather than palliative intent.

- Table 1: I believe that the main conclusions must be summarized and presented in topics rather than text. This column must appear sooner in the table and before the “signalling pathway” column.

R: We thank the reviewer for the suggestions. However, we partially agree with the comments. We summarize the findings and present them in bullet points, but we think this column should remain where it is.

- Table 1: rename the column “signalling pathway”, because it actually refers to “genes”.

R: The column has been renamed.

We included bibliographic references in table 1.

- include references at section 3.3.2.            

R: We didn't understand this issue. Please enlighten us.

- figures 1, 2 and 3 must be improved: font (same as the manuscript), colour (black legend), remove the superior legend. Remove “series 1 and 2”.

R: All suggestions were accepted and corrected in the manuscript.

- figures 1-3 must be mentioned in the text.

R: All figures were mentioned in the text.

  1. Discussion

- Please include references in ALL information that is not yours! Otherwise, is considered plagiarism.

R: We include references in all information.

- Discussion must be re-written. I believe that after performing the changes suggested in these comments and after digesting further the results, it would be easier to write the discussion.

R: We thank the reviewer for critical suggestions that significantly improved our manuscript. We made substantial changes to the discussion. All changes are highlighted in yellow.

Reviewer 3 Report

Comments and Suggestions for Authors

The revised version is great.

Comments on the Quality of English Language

The revised version is great.

Author Response

Reviewer 3

The revised version is great.

R: We greatly appreciate the reviewer comment to our manuscript. 

Round 2

Reviewer 2 Report

Comments and Suggestions for Authors

The manuscript was significantly improved. I think it can be publish in the present form.